# *Paracoccidioides lutzii* Formamidase Contributes to Fungal Survival in Macrophages

**DOI:** 10.3390/microorganisms10102011

**Published:** 2022-10-12

**Authors:** Lana O’Hara Souza Silva, Thalison Rodrigues Moreira, Relber Aguiar Gonçales, Mariana Vieira Tomazett, Juliana Alves Parente-Rocha, Karine Mattos, Juliano Domiraci Paccez, Orville Hernandez Ruiz, Maristela Pereira, Célia Maria de Almeida Soares, Simone Schneider Weber, Vanessa Rafaela Milhomem Cruz-Leite, Clayton Luiz Borges

**Affiliations:** 1Laboratory of Molecular Biology, Institute of Biological Sciences, Federal University of Goiás, Goiânia 74690-900, Brazil; 2Life and Health Sciences Research Institute (ICVS), School of Medicine, University of Minho, 4700-000 Braga, Portugal; 3ICVS/3B’s—PT Government Associate Laboratory, 4800-000 Braga, Portugal; 4Bioscience Laboratory, Faculty of Pharmaceutical Sciences, Food and Nutrition, Federal University of Mato Grosso do Sul, Campo Grande 79070-900, Brazil; 5MICROBA Research Group—Cellular and Molecular Biology Unit—CIB, School of Microbiology, University of Antioquia, Medellín 050010, Colombia

**Keywords:** nitrogen catabolite repression, gene silencing, nitrogen depletion, macrophage infection, virulence

## Abstract

Nitrogen is a crucial nutrient for microorganisms that compose essential biomolecules. However, hosts limit this nutrient as a strategy to counter infections, therefore, pathogens use adaptive mechanisms to uptake nitrogen from alternative sources. In fungi, nitrogen catabolite repression (NCR) activates transcription factors to acquire nitrogen from alternative sources when preferential sources are absent. Formamidase has been related to nitrogen depletion in *Aspergillus nidulans* through formamide degradation to use the released ammonia as a nitrogen source. In *Paracoccidioides* spp., formamidase is highly expressed in transcriptomic and proteomic analyses. Here, we aim to investigate the importance of formamidase to *Paracoccidioides lutzii*. Thereby, we developed a *P. lutzii* silenced strain of *fmd* gene (AsFmd) by antisense RNA technology using *Agrobacterium tumefaciens*-mediated transformation (ATMT). The AsFmd strain led to increased urease expression, an enzyme related to nitrogen assimilation in other fungi, suggesting that *P. lutzii* might explore urease as an alternative route for ammonia metabolism as a nitrogen source. Moreover, formamidase was important for fungal survival inside macrophages, as fungal recovery after macrophage infection was lower in AsFmd compared to wild-type (WT) strain. Our findings suggest potential alternatives of nitrogen acquisition regulation in *P. lutzii*, evidencing formamidase influence in fungal virulence.

## 1. Introduction

The genus *Paracoccidioides* is a complex of thermally dimorphic fungi composed by five species: *Paracoccidioides brasiliensis*, *Paracoccidioides lutzii*, *Paracoccidioides americana*, *Paracoccidioides restrepiensis* and *Paracoccidioides venezuelensis* [1]. Recently, after phylogenic and population genetic analysis, two uncultivated fungi were identified and proposed as new *Paracoccidioides* species: *Paracoccidioides cetii*, a dolphin pathogen, and *Paracoccidioides loboi*, a human pathogen [2]. These are the etiological agents of Paracoccidioidomycosis (PCM), a human disease widely distributed throughout Latin America and one of the most important systemic mycoses at this region [3,4]. Infection occurs through inhalation of infectious propagules produced by the saprobiotic mycelial form, found in soil. These mycelia propagules reach the host’s lungs and germinate into pathogenic yeast form due to contact with the host milieu, and driven by body temperature [5].

The first contact of *Paracoccidioides* yeast cells with the host’s immunological system occurs with alveolar macrophages, which promote granuloma formation to contain fungal dissemination. This component of innate immunity possesses a harsh microenvironment that impairs fungal survival with factors such as acid pH, oxidative stress and nutrient limitation. Hence, the pathogen must develop strategies to counter macrophage attacks to promote infection/dissemination [5,6,7]. *Paracoccidioides* spp. is a facultative intracellular pathogen, so it is able to survive and replicate within macrophages, which indicates that this pathogen has developed mechanisms, such as the production of adhesins and antioxidants, to overcome the hostile condition found in the host microenvironment [8,9,10]. Some of these molecules produced by the *Paracoccidioides* genus during infection can contribute to fungi pathogenesis and, therefore, are denominated virulence factors/determinants. The main virulence determinants produced by *Paracoccidioides* spp. include the glycoprotein 43 kDa (Gp43), glycoprotein Gp70 kDa, α-(1,3)-glucan, glyceraldehyde-3-phosphate dehydrogenase (GAPDH) and melanin, which are involved in processes such as adherence, response to stress conditions and confronting host innate immune components [10,11]. Other virulence determinants of the *Paracoccidioides* genus and their function on fungal pathogenesis were reviewed by [11]. Another crucial aspect during infection is nutrient acquisition. To establish the infection, the microorganism needs to acquire nutrients from the host that are crucial for biological processes, such as carbon and nitrogen. Carbon is the main source of energy metabolism, whereas nitrogen is mainly necessary for protein and acid nucleic biosynthesis [12]. Nonetheless, these nutrients are not completely available inside the host, especially when the pathogen is phagocyted by macrophages, which have an extremely nutrient-depleted microenvironment. Therefore, the pathogen must adapt to this condition and scavenge nutrients from alternative sources, before reaching a nutrient-rich environment, such as the bloodstream [13].

Nitrogen is one of the most important nutrients for microorganisms because it is essential for the synthesis of crucial molecules, such as proteins and nucleic acids. When this nutrient is scarce, fungal cells reduce their growth. In the case of extremely low nitrogen availability, cell growth can be ceased, even in the presence of other important nutrients [12]. There is a broad range of nitrogenous compounds that can be used as nitrogen sources by fungi. Some sources can be easily assimilated and are preferentially used, such as ammonia, glutamine and glutamate. When preferential sources become unavailable due to host immune system action, the pathogen needs to uptake nitrogen from less preferred sources, such as nitrates, nitrites, proteins and some amino acids. In fungi, nitrogen acquisition is regulated by the nitrogen catabolic repression (NCR) mechanism. This mechanism prioritizes the utilization of preferred nitrogen sources when available, repressing permeases and enzymes codified by genes involved in the catabolism of secondary sources. Otherwise, in the absence of preferred nitrogen sources, NCR transcription factors derepress the NCR-sensitive genes implicated in nitrogen assimilation from alternative sources [14,15].

During NCR, numerous genes are under regulation in order to acquire nitrogen from secondary sources. Enzymes whose expression is regulated during NCR have been targets in several studies. One example is the formamide amidohydrolase (formamidase, Fmd, EC 3.5.1.49), which hydrolyzes formamide to produce formate and ammonia. In *Aspergillus nidulans*, this enzyme contributes to nitrogen acquisition, releasing ammonia through formamide degradation. Furthermore, there is a GATA motif sequence on the promoter region of *fmdS*, where the main *A. nidulans* NCR transcription factor, AreA, binds and regulates gene expression. The absence of this region prevents *fmdS* upregulation during nitrogen starvation [16]. In *P. lutzii*, formamidase was upregulated after fungal growth in NCR conditions, where proline was used as a non-preferential nitrogen source, suggesting a role in nitrogen assimilation (Cruz-Leite, unpublished data). Moreover, similar to *A. nidulans*, the formamidase gene of *P. lutzii*, *P. brasiliensis* and *P. americana* possess the GATA motif sequence in the promoter region, implicating the regulation of formamidase gene by a GATA family transcription factor such as AreA [17].

Besides the putative involvement in nitrogen metabolism in *Paracoccidioides* spp., formamidase is also thought to act as a virulence determinant. The formamidase gene was the second most abundant on transcriptome analysis of *P. brasiliensis* [18], and one of the most abundant in *P. brasiliensis* yeast cells recovered from livers of infected mice [19]. It was recognized with strong reactivity in sera of patients with paracoccidioidomycosis, implying its association with fungal pathogenesis [20]. Furthermore, this enzyme is located in the fungal cytoplasm and cell wall and interacts with other proteins also present on the cell wall, suggesting its involvement in fungal antigenic properties [21,22]. Corroborating this, formamidase was identified as an exoantigen of *P. lutzii* [23], and was upregulated during ex vivo [24,25] and in vivo [26] experimental infection with *P. brasiliensis.* Additionally, formamidase was the most abundant on proteome analysis of *P. lutzii* conidia [27] and was upregulated during *P. brasiliensis* mycelium-to-yeast transition [28], suggesting a role of this enzyme during the first steps of infection with the *Paracoccidioides* species.

The acquisition of carbon and micronutrients by *Paracoccidioides* has been widely studied by our group and is well defined (reviewed in [29]); however, little is known about nitrogen acquisition by this pathogenic fungus. In a previous work, we identified the NCR mechanisms involved with nitrogen non-preferential sources in the *Paracoccidioides* species [17] and the upregulation of formamidase during nitrogen depletion (Cruz-Leite, unpublished data). In order to evaluate formamidase function in *P. lutzii*, we silenced the *Plfmd* gene through antisense RNA technology mediated by *Agrobacterium tumefaciens* (ATMT). We observed that the silencing of the formamidase coding gene impaired fungal survival within macrophages, suggesting a role for this enzyme as a virulence determinant in *P. lutzii*. Additionally, low levels of formamidase enzyme in the silenced strain increased urease expression and activity, suggesting that *P. lutzii* can use urease as an alternative route for ammonia production and subsequent assimilation as a nitrogen source. To our knowledge, this is the first work applying RNA-antisense silencing technology in *P. lutzii* cells.

## 2. Materials and Methods

### 2.1. Ethics Statement

Experiments with mice were performed in accordance with the ethical principles of animal research adopted by the Brazilian Society of Laboratory Animal Science and the Brazilian Federal Law 11.749/2008. Animal experimentation was approved by the Ethics Committee on the use of animals of the Federal University of Goiás (reference number 093/16). All efforts were made to minimize animal suffering.

### 2.2. Fungal Strains and Growth Conditions

*P. lutzii* (ATCC MYA-826) wild-type (WT) yeast cells and *P. lutzii fmd* silenced strains (AsFmd) were used in all experiments. Cells were maintained in solid brain-heart infusion (BHI) media, supplemented with 1.1 % (*w*/*v*) glucose, at 37 °C for 3 or 5 days. For liquid cultures in BHI, cells were submitted to continuous shaking (150 rpm) at 37 °C for 72 h until they reached the exponential growth phase.

### 2.3. Construction of fmd P. lutzii Silenced Strains with Antisense RNA Technology

The *P. lutzii* formamidase gene (*fmd*) was silenced through the antisense RNA (aRNA) technique along with ATMT technology, as described previously [24,30,31]. Briefly, the oligonucleotides F (5′ CCGCTCGAGCGGCTTGCATAACCGCTGGCATC 3′) and R (5′ GGCGCGCCTCGTCGGCGGAATCGTTATT 3′) were designed to generate an antisense fragment (123 bp) of the *P. lutzii fmd* gene (PAAG_03333; Accession number obtained in the FungiDB database at https://fungidb.org/fungidb/, accessed on 19 July 2022). The *Plfmd* antisense fragment was amplified by PCR and cloned into the pCR35 plasmid, which harbors the promoter region of the *Histoplasma capsulatum cbp1* gene and the terminating region of the *Aspergillus fumigatus cat-B* gene [32]. The plasmid pUR5750, which contains a transfer-DNA (T-DNA) harboring the hygromycin (Hyg)-resistance gene (*hph)* from *Escherichia coli*, was used as a binary vector to bear the *Plfmd*-aRNA cassette. *Agrobacterium tumefaciens* (LBA1100) cells were transformed with the cassette, cultured in induction medium (IM) [containing spectinomycin (250 µg/mL), kanamycin (100 µg/mL), rifampicin (20 µg/mL) and acetosyringone 0.4 M] at 28 °C, and co-cultivated with *P. lutzii* (1:10 ratio) in sterile Hybond^TM^ N+ membrane (GE Healthcare, Little Chalfont, UK) on solid IM at 28 °C for 3 days. Following co-cultivation, *P. lutzii* silenced cells were recovered as described by Almeida et al. [30] and mitotic stability was evaluated into a selective BHI [1.1% (*w*/*v*) glucose] solid medium containing hygromycin at 75 µg/mL for three passages, and then alternating the absence and presence of hygromycin 75 µg/mL for seven consecutive passages. This analysis was performed in experimental triplicate. Afterward, fungal strains were maintained in the selective BHI medium containing hygromycin 75 µg/mL.

### 2.4. Validation of P. lutzii AsFmd Transformants and Expression Analysis by RT-qPCR

AsFmd transformants were selected and confirmed through resistance to hygromycin. *P. lutzii* cells were collected after 72 h of growth in BHI liquid media at 37 °C with constant shaking (150 rpm), and were seeded on non-selective BHI and selective BHI containing hygromycin at 75 µg/mL [1.1 % (*w*/*v*) glucose] at 10^5^ and 10^4^ cells/mL dilutions. Plates were incubated for 7 days at 37 °C. Hygromycin-resistant transformants were selected to confirm pUR5750::*Plfmd*-aRNA integration and to evaluate the number of insertional events.

For DNA extraction, yeast *P. lutzii* cells were collected from BHI solid media [1.1% (*w*/*v*) glucose] under selective pressure of hygromycin 75 µg/mL supplementation after 72 h of growth at 37 °C. Cells were resuspended in STES buffer [0.2 M Tris-HCl pH 7.2, 0.5 M NaCl, 10 mM EDTA, 0.1% (*w*/*v*) SDS] supplemented with RNAse (1 μg/mL), lysed through mechanical disruption (Mini-Beadbeater—Biospec Products Inc., Bartlesville, OK, USA) and DNA was extracted according to standard procedures [33]. Thus, T-DNA cassette integration was confirmed through conventional PCR with oligonucleotides designed to amplify the integration region: pUR5750 F (5′ GGAAAGCCGGCGAACGTGG 3′) and pUR5750 R (5′ CAGATTCGAAAGCGCCTTCAG 3′).

For RNA extraction, yeast *P. lutzii* cells were incubated for 48 h in BHI liquid media [1.1% (*w*/*v*) glucose] and constant shaking at 150 rpm. Next, RNA was extracted with TRIzol™ Reagent (TRI reagent, Sigma Aldrich, St. Louis, MO, USA) and mechanical cell rupture (Mini-Beadbeater—Biospec Products Inc., Bartlesville, OK, USA). After that, RNA was reverse-transcribed using the M-MLV Reverse Transcriptase (Sigma Aldrich). The cDNA generated was submitted to quantitative PCR (qPCR) with SYBR™ Green PCR master mix in a QuantStudio 5 Real-Time PCR System (Applied Biosystems, Foster City, CA, USA). Transcriptional levels of the exogenous hygromycin gene were accessed using the oligonucleotides designed *hph* F: (5′ CACCTCGTGCACGCGGATTTCG 3′) and *hph* R: (5′ CCAACCACGGCCTCCAGAAGAAGA 3′). Data was normalized with the oligonucleotides designed for enolase coding gene (PAAG_11169; FungiDB database accession number at https://fungidb.org/fungidb/, accessed on 19 July 2022), *eno* F: (5′ GATTTGCAGGTTGTCGCCGA 3′), and *eno* R: (5′ TGGCTGCCTGGATGGATTCA 3′), and qPCR was performed with biological triplicates in each reaction. Relative expression levels of transcripts were calculated using a standard curve at serial dilution of 1:5, for relative quantification [34]. Statistical analysis was performed by the ANOVA, and *p*-values ≤ 0.05 were considered statistically significant. The total DNA hygromycin copy number was determined with the standard curve method (Cts plotted against logarithm of DNA copy number), and the calibrator for copy number of hygromycin random integrations was determined by [35]. Results were expressed as N-fold changes in target gene copies normalized with the enolase reference gene. For N-fold values that ranged from 0.7 to 1.3, the transformant strains were considered to harbor a single integrated copy of the hygromycin gene [36]. Strains pENO.5:*Pl* [35] and *P. falciparum* Dd2 [36] containing known copy numbers were used to estimate the copy number in the silenced mutants. Linear regression analysis and statistical analysis using ANOVA method were performed using Excel software with 99% confidence. R and standard error were evaluated.

Transcriptional levels of the *P. lutzii fmd* gene (PAAG_03333; FungiDB database accession number at https://fungidb.org/fungidb/, accessed on 19 July 2022) and the *P. lutzii* urease gene (PAAG_00954; FungiDB database accession number at https://fungidb.org/fungidb/, accessed on 19 July 2022) from WT and AsFmd strains were accessed using the oligonucleotides designed for the *fmd* gene: *fmd* F (5′ GTTCTATCCCAATGCCGCAAA 3′) and *fmd* R (5′ GCGCCTGTTCCATTCAGCTA 3′), and for the urease gene: *ure* F (5′ GAGATATATGTTTGGGGCACG 3′) and *ure* R (5′ ACCTCGACAATTACGCACCG 3′). Data was normalized with the oligonucleotides designed for the tubulin coding gene (PAAG_03031; FungiDB database accession number at https://fungidb.org/fungidb/, accessed on 19 July 2022): *tub* F (5′ GATAACGAGGCTCTGTATGATA 3′) and *tub* R 5′ (ATGTTGACGGCGAGTTTGCG 3′), as previously described [17,24]. qPCR was performed with biological triplicates and relative expression levels of transcripts were calculated using a standard curve for relative quantification [34]. Statistical analysis was performed by Student’s t-test and *p*-values ≤ 0.05 were considered statistically significant.

### 2.5. Growth Curve and Viability of P. lutzii Silenced Cells

*P. lutzii* WT and AsFmd yeast cells were cultured in liquid BHI [1.1% (*w*/*v*) glucose] with constant shaking at 37 °C. To determine the growth curve, optical density was measured in triplicate, every 24 h during 96 h, at a wavelength of 600 nm. Viability was assessed through staining with 1 μg/mL (*w*/*v*) propidium iodide (Sigma Aldrich) and samples were visualized in a fluorescence microscope at 493/636 nm (Axioscope A1—Carl Zeiss AG, Jena, Germany) [37].

### 2.6. Total Protein Extraction and Fmd Immunoblotting Assay

In order to obtain the total protein extract, WT and AsFmd yeast cells were cultivated in liquid BHI with constant shaking at 37 °C for 72 h. Afterward, cells were harvested and lysed through a mechanical rupture in the presence of 50 mM ammonium bicarbonate pH 8.5 solution. Protein concentration was determined using a Bradford assay [38]. For immunoblotting assay, a total of 30 μg of protein extract from WT and AsFmd strains was loaded on 12% SDS-PAGE and separated by electrophoresis. After that, proteins were transferred to the nitrocellulose membrane, and complete protein transfer was checked with Ponceau red staining. Non-specific sites were blocked after incubation with a blocking buffer [Phosphate buffered saline (PBS), 5% (*w*/*v*) non-fat milk and 0.1% (*v*/*v*) Tween 20] for 2 h. Afterwards, membranes were washed with a wash buffer [Phosphate-buffered saline (PBS), 0.1% (*v*/*v*) Tween 20] and incubated with anti-*Pb*Fmd polyclonal antibodies (diluted 1:500), for 1 h 30 min (adapted from Borges et al. [22]). Membranes were washed with a wash buffer and incubated with the secondary antibody anti-mouse IgG, coupled to peroxidase, (diluted 1:1000), from the Pierce ECL Western Blotting Substrate Kit (Promega, Madson, WI, USA). The reaction was developed in a chemioluminescent imager (Amersham Imager 600, GE Healthcare). Three independent experiment replicates were performed.

### 2.7. Measurement of Formamidase and Urease Activity

Protein extracts of WT and AsFmd were used to measure formamidase and urease enzymatic activity. Formamidase and urease activity were measured by monitoring ammonia release, as described in [20,22], with modifications. Briefly, 500 ng of protein extracts were added to 50 μL of 100 mM formamide (for formamidase activity) or 50 mM urea (for urease activity) substrate solution in 100 mM phosphate buffer, pH 7.4, and 10 mM EDTA. The reaction mixture was incubated at 37 °C for 30 min. Next, 80 μL of phenol-nitroprusside and 80 μL alkaline hypochlorite (Sigma Aldrich) were added to the reaction and samples were incubated at 50 °C for 6 min. Absorbance was read at 625 nm and the amount of ammonia released was calculated compared with a standard curve. One unit (U) of formamidase and urease activity was defined as the amount of enzyme required to hydrolyze 1 μmol of formamide and urea, respectively, per minute per mg of total protein. Three experimental replicates were performed.

### 2.8. Urease Heterologous Expression

The urease coding gene, *ure*, of *P. brasiliensis* (PADG_03871; FungiDB database accession number at https://fungidb.org/fungidb/, accessed on 19 July 2022) was amplified from cDNA with the primers *Pbure* F (5′ GGATCCATGGCTCTAGGCAAGACC 3′) and *Pbure* R (5′ GGATCCTCAATAGACAAAATACTGCTG 3′) and subcloned into the *Bam*HI site of pET32a vector (Novagen). The obtained plasmid (pET32a::*Pbure*) was transformed into *E. coli* pLysS. Prior to protein induction, transformed *E. coli* cells were cultured in LB (Luria Bertani) medium containing 100 μg/mL ampicillin for 16 h at 37 °C. Urease was induced by addition of 1.0 mM of isopropyl β-D-1-thiogalactopyranoside (IPTG; Sigma-Aldrich) for 3 h. Recombinant urease (rUre) was solubilized with 1 mg/mL lysozyme for 1 h followed by incubation with 1% sarcozil and sonication with an ultrasonic bath (Ultrassonic Cleaner—Odontobras), with 5 cycles of 10 min and 30 s intervals in vigorous shaking (VX-200 Vortex mixer—Labnet). The samples were passed 10 times in a 0.8 mm needle and 10 times in a 0.45 mm needle and centrifuged for 20 min at 16,000× *g.* The soluble fraction was purified by affinity chromatography after incubation in a NI-NTA AGAROSE resin (Invitrogen^TM^, Waltham, MA, USA) and elution with 250 mM imidazol. rUre size and identity was evaluated by SDS-PAGE followed by mass spectrometry analysis.

### 2.9. Ure Polyclonal Antibodies Production and Immunoblotting

rUre was utilized for polyclonal antibody production in *BALB*/*c* mice. Mice were subjected to three subcutaneous injections at 15 days intervals: the first with 100 μg of purified protein in Freund’s Complete Adjuvant (Sigma-Aldrich) and the subsequent two with 100 μg of purified protein in Freund’s Incomplete Adjuvant (Sigma-Aldrich). The pre-immune serum containing polyclonal antibodies anti-*Pb*Ure were collected and stored at −20 °C.

A total of 30 μg of protein extracts from *P. lutzii* WT and *AsFmd* strains, and 10 μg of purified rUre, was loaded and separated on 12% SDS-PAGE. Immunoblotting was carried out as described previously (item 2.7). Anti-*Pb*Ure polyclonal antibodies were used as the primary antibody at the dilution of 1:5000. The antibody anti-mouse IgG, coupled to peroxidase, (diluted 1:1000), from Pierce ECL Western Blotting Substrate Kit (Promega), was used as the secondary antibody. The reaction was developed in a chemioluminescent imager (Amersham Imager 600, GE Healthcare). This experiment was carried out in experimental triplicate.

### 2.10. Ex-Vivo Infection of AsFmd in J774 Macrophages

The abilities of internalization and survival inside macrophages of *P. lutzii fmd* silenced strains was carried out through ex vivo infection assay. Murine macrophages from the cell line J774 1.6 (Rio de Janeiro Cell Bank—BCRJ/ UFRJ, accession number 0273) were used in this experiment. They were maintained in Dulbecco’s Modified Eagle Medium (DMEM) High Glucose (Sigma-Aldrich), containing 10% glucose (*v*/*v*) at 37 °C and 5% of CO_2_. Meanwhile, *P. lutzii* cells (WT and AsFmd) were maintained in BHI filled with glucose 1.1% (*v*/*v*) at 37 °C and constant shaking of 150 rpm for 48 h. Next, the macrophage cells were harvested and an amount of 10^6^ cells/mL were applicated on a polypropylene 12-well-plate and primed using 100 U/mL of IFN-ƴ (Biolegend, San Diego, CA, USA) in DMEM for 16 h. *P. lutzii* WT and AsFmd strains were independently added to a 12-wells-plate containing a rate of 5:1 yeast:macrophage cells and incubated for 24 h at 37 °C and 5% of CO_2_. After the incubation period, the wells were rinsed twice with a sterile saline buffer in order to remove the cells non-internalized/adhered by macrophages. Next, ice-cold sterile water was used to disrupt macrophage membranes and collect *P. lutzii* cells. Recovered *P. lutzii* cells were streaked on solid BHI medium supplemented with glucose 2% (*v*/*v*) and fetal bovine serum 10% (*v*/*v*) and incubated at 37 °C for seven days, when the colony-forming units (CFU) were performed. CFUs were expressed as the mean value ± the standard error from experimental duplicates of two biological replicates and Student’s *t*-test was applied for statistical analyses.

### 2.11. Statistical Analysis

First, the population distribution was investigated through the Shapiro–Wilk test. The hypothesis of non-normality of the data was rejected at *p*-value ≤ 0.05. Next, Student’s t-test was performed in the following experiments: RT-qPCR (*fmd* and *ure*), enzymatic activity, immunoblotting analyses and *ex vivo* infection. Data were considered statistically significant at *p* value ≤ 0.05. For RT-qPCR (*hph*), statistical analysis was performed through one way ANOVA and *p*-values ≤ 0.05 were considered statistically significant. All assays were performed in experimental triplicates, except for macrophage infection, which was carried out in biological duplicate and experimental triplicates. GraphPad Prism 8.0.1 software (San Diego, CA, USA) and Excel software were used to run the statistical analyses.

## 3. Results

### 3.1. Production of Fmd P. lutzii Silenced Strains

Silenced *fmd P. lutzii* strains were generated through ATMT technology. The T-DNA cassette containing AsFmd, was randomly integrated into *P. lutzii* yeast cells (Figure 1a). T-DNA cassette integration into the *P. lutzii* genome was confirmed through conventional PCR using pUR5750 oligonucleotides. Three transformed colonies were randomly selected for analysis. An amplicon of 1800 bp was observed only in the transformed cells, confirming the insertion of T-DNA cassette (Figure 1b). To evaluate the hygromycin expression level and gene copy number, we conducted a qPCR assay in *P. lutzii* AsFmd transformants strains. To determine N-fold values as described by [36], threshold cycle (Ct) values of hygromycin were compared with a reference strain pENO.5:Pl containing the single copy integration [35]. Published data from *P. falciparum* Dd2 [36] was also used for comparison and the estimative of copy number in AsFmd15 was accessed through linear regression analysis. *P. lutzii* AsFmd4 and AsFmd14 transformants presented N-fold values ranging from 0.7 to 1.3, indicating one single copy integration. On the other hand, AsFmd15 transformant presented a 1.76 N-fold value, indicating the presence of two integrations (Figure 1c). Furthermore, as demonstrated in Figure 1c, the transcriptional hygromycin levels were not detected in wild-type strains, whereas in transformants AsFmd4, AsFmd14 and AsFmd15 the levels were significant. These positive strains were then analyzed for resistance to hygromycin (*hph* gene), a resistant marker present on T-DNA (Figure 1a). WT and AsFmd strains were grown in BHI or BHI + Hyg^75^. Consistent with PCR results, all transformed strains were able to grow in the presence of hygromycin and the WT strain had an impaired growth when this antibiotic was present (Figure 1d).

### 3.2. Reduced Plfmd Gene Expression Does Not Impair Fungal Growth Rate and Cell Viability

The *Pl*fmd transcriptional level was analyzed in WT and all positive AsFmd strains through RT-qPCR. The silencing percentage of AsFmd strains ranged from 64% to 80% (Figure 2a). The silenced *P. lutzii* strain (AsFmd15) that exhibited 80% of reduced formamidase transcripts was selected for further experiments. Similarly, *Pl*Fmd protein expression was also reduced 6-fold in the mutant strain when compared with WT, as seen in immunoblotting analysis in the expression of a 45-kDa protein equivalent to *P. lutzii* formamidase [20] (Figure 2b). The growth and viability were evaluated in BHI and with propidium iodide staining, in the WT as well as in the AsFmd strains. The knock-down of *fmd* did not impair fungal growth (growth curve) nor cell viability (microscopy) (Figure 2c), indicating that the *fmd* silencing process is not detrimental for *P. lutzii* yeast cells in regular growth conditions. In addition to reduced protein expression, the AsFmd strain also exhibited reduced (0.75-fold) Fmd activity when compared to the WT strain (Figure 2d), demonstrating that the silencing process not only reduced Fmd levels but also its activity.

### 3.3. Decreased PlFmd Expression Levels Influence PlUre Expression and Activity

Urease and formamidase can be fungal sources of ammonia release. While formamidase releases ammonia through the degradation of formamide, urease hydrolyzes urea to release it [16,39]. The ammonia released by urease activity can contribute to pathogenesis of bacteria and fungi [40,41], as well as serving as a nitrogen source [42,43]. Therefore, we evaluated if urease expression and enzymatic activity were affected by low levels of formamidase. First, we analyzed urease expression through RT-qPCR and observed a significant increase (1.2-fold) of urease expression in the AsFmd strain, compared to WT (Figure 3a). After this result, we decided to analyze urease expression at the protein level. For this, *P. brasiliensis* urease was successfully expressed in a bacterial system, purified and used for polyclonal antibody production in mice (Figure 3b). Antibody anti-*Pb*Ure specificity was tested with purified rUre and identified a protein with molecular mass of 103 kDa, corresponding to 83 kDa of *Pb*Ure fused to 20 kDa of His-tag (Figure 3c). Next, we tested if anti-*Pb*Ure was capable of recognizing urease native protein in *P. lutzii*, since it was produced using the *P. brasiliensis* urease sequence as reference. These proteins share 98% similarity and 98% identity (data not shown). A protein of 85 kDa, corresponding to *P. lutzii* urease molecular mass, was identified (Figure 3c). These results indicate that the anti-*Pb*Ure antibody is specific and able to recognize both the recombinant and the native *P. lutzii* urease. Next, we analyzed urease expression and enzymatic activity in the WT and AsFmd strains. As seen in the immunoblotting analysis, urease expression was 5.2-fold higher in the *fmd* silenced strain when compared with WT (Figure 3d). In agreement, urease activity was significantly increased (1.4-fold) in the *fmd* silenced strain, compared to WT (Figure 3e). With these findings, we can conclude that urease expression and enzymatic activity are increased at reduced formamidase expression in the silenced strain. The results demonstrate the expression and activity of urease increase during the reduced formamidase expression in the *P. lutzii* AsFmd strain.

### 3.4. PlFmd Is Important for P. lutzii Survival during Macrophage Infection

In previous works, formamidase was associated with several mechanisms that might be related to pathogenicity in the Paracoccidioides species. However, it is unclear if formamidase can influence the virulence of the Paracoccidioides spp. To evaluate formamidase’s role in *P. lutzii* internalization and survival within macrophages, we performed macrophage infection with the WT and AsFmd strains. Fungal survival was analyzed by CFU counting. Student’s t-test showed a significant difference when the CFU of WT and AsFmd were compared. Additionally, an ample *p* value was generated, indicating elevated dissimilarity surrounding the growing profile of infection-recovered fungal cells between the groups referred to (Figure 4). Therefore, these results suggest a role played by Fmd in the *P. lutzii* response to the nutritional depletion strategy imposed by alveolar macrophages in first contact with a pathogen’s cells placed in surfactant film inside the alveoli.

## 4. Discussion

During host–pathogen interactions, several genes are regulated to promote fungal survival and virulence to favor infection. Most metabolic adjustments executed by pathogens during infection are related to cell defense, adhesion, and melanin production, as well as nutrient acquisition [13,44], especially for pathogens such as *Paracoccidioides* spp., which is able to replicate within macrophages. These immunological cells possess a hostile microenvironment with low nutrient availability, oxidative stress and acidic pH that impairs fungal survival, which indicates that *Paracoccidioides* spp. has developed mechanisms to circumvent these harsh conditions and promote infection [6,45]. This highlights the importance of investigating the genes regulated during infection and identifying possible virulence determinants.

In *P. brasiliensis*, formamidase is one of the most induced (up-regulated) enzymes during infection [24,25,26]. Additionally, formamidase reacts with the sera of patients with PCM [20] and was identified as an exoantigen of *P. lutzii* [23]. Formamidase is also related to nitrogen metabolism, as its expression is regulated according to nitrogen availability in *A. nidulans* [16], and was up-regulated after the growth of *P. lutzii* in proline as the sole nitrogen source (Cruz-Leite, unpublished data). Since formamidase has been associated with virulence determinants and nitrogen metabolism in the *Paracoccidioides* species, we silenced the *Plfmd* gene through antisense RNA technology associated with ATMT to elucidate its biological role in *P. lutzii*. This technique has contributed to the identification of proteins required for relevant processes of *P. brasiliensis* pathobiology, such as Cdc42 [46], HSP90 [47], Rbt5 [48], Ccp [24], PCN [49], SidA [37], FglA [50] and HSP30 [31].

In this work, this method was applied for the first time in *P. lutzii* cells. We successfully obtained knocked-down strains for *Plfmd*, as demonstrated by RT-qPCR and immunoblotting and enzymatic activity. Silencing of the *Plfmd* gene did not impair fungal growth or viability under regular cultivation conditions, indicating that the *fmd* silencing process does not influence fungal in vitro growth or viability. This finding does not eliminate the suspect role of formamidase as a virulence determinant in *P. lutzii*. As described by Rappleye and Goldman [44], a virulence determinant for dimorphic fungi is defined as being required for pathogen growth and survival during infection in its mammalian host, but not for growth during in vitro assay. Corroborating this, we observed a reduction in fungal survival after macrophage infection in the *fmd* silenced strains, suggesting that formamidase might be important to *P. lutzii* survival during infection.

The silencing of *fmd* resulted in an increase of urease expression. Urease (EC 3.5.1.5; urea amidohydrolase) is a metalloenzyme that hydrolyzes urea to yield carbon dioxide and ammonia [39]. During infection, the ammonia released from urea hydrolysis can be a virulence determinant, causing damage to the host tissue or helping to counteract the acidic pH within phagosomes [43,51], or can be used as a nitrogen source [42,43]. The influence of formamidase on urease expression was not observed in any organism so far. The known relation between urease and formamidase is that urease is able to hydrolyze formamide, but as a poor substrate [52]. In *Helicobacter pylori*, formamide hydrolysis was diminished, but not abolished, in a formamidase-deleted strain. Abolishment of formamide hydrolysis was only achieved when both formamidase and urease were deleted, indicating that formamide hydrolysis is dependent on formamidase and urease activity in this bacteria [53]. Urease activity was induced during formamide limitation in the methylotrophic bacterium *Methylophilus methylotrophus* [54].

In *P. brasiliensis*, an increase of extracellular pH was observed during the late-log phase of growth, probably related to urease activity [55]. Additionally, cultivation of *P. lutzii* under low nitrogen availability up-regulated a putative urease accessory protein, UreG (Cruz-Leite, unpublished data), which was described as being important for urease activation in bacteria [56,57]. These data suggest that urease might play a role in nitrogen acquisition and in extracellular alkalinization, in *Paracoccidioides* species. Our finding of the influence of formamidase on urease expression and activity in *P. lutzii* suggests that, at low levels of formamidase, urease is increased as an alternative route for ammonia release. However, the real role of urease in this condition remains unknown. Either urease is being up-regulated to degrade urea and use the ammonia released as a nitrogen source, or to act as an alternative pathway for formamide hydrolysis. Taking into account that urease is probably covering the low levels of formamidase, this could be one of the explanations for why the silencing of *fmd* did not impair *P. lutzii* growth or viability. In this sense, it is necessary for further experiments to elucidate formamidase and urease relationships in *P. lutzii*.

A previous study with *P. brasiliensis* raised the concept of formamidase being a virulence factor. They observed that *fmd* transcripts were increased in a virulent isolate (isolated from mice) compared to an attenuated isolate (continuous in vitro cultivation) [58]. In the present study, we found that *P. lutzii* formamidase might be essential for fungal survival within macrophages, as *fmd* silencing reduced fungal recovery after macrophage infection. This finding reinforces the role of formamidase in host–fungus interactions. Macrophages are the first line of defense recruited by the host immune system during infection and present a hostile microenvironment to counter the pathogen. Being able to overcome and survive the action of this component of innate immunity is a virulence determinant essential for establishment of the infection [6]. Therefore, our data support the concept of formamidase being a virulence determinant in *P. lutzii*.

## 5. Conclusions

Through silencing with antisense RNA technology, we were able to investigate formamidase function in *P. lutzii*. Formamidase probably plays a role during nitrogen acquisition, since reduced levels of this enzyme increased expression and activity of another enzyme, urease, that also releases ammonia during its activity. The silencing of formamidase impaired fungal survival during macrophage infection, indicating a possible role of this enzyme as a virulence factor for *P. lutzii*. Additional studies are needed to determine formamidase and urease correlation in *P. lutzii* nitrogen metabolism. Furthermore, complementary work is necessary to establish formamidase as a virulence determinant of *P. lutzii*, and hence, to elucidate the adaptation strategies promoted by the pathogen during host–pathogen interactions.

## Figures and Tables

**Figure 1 microorganisms-10-02011-f001:**
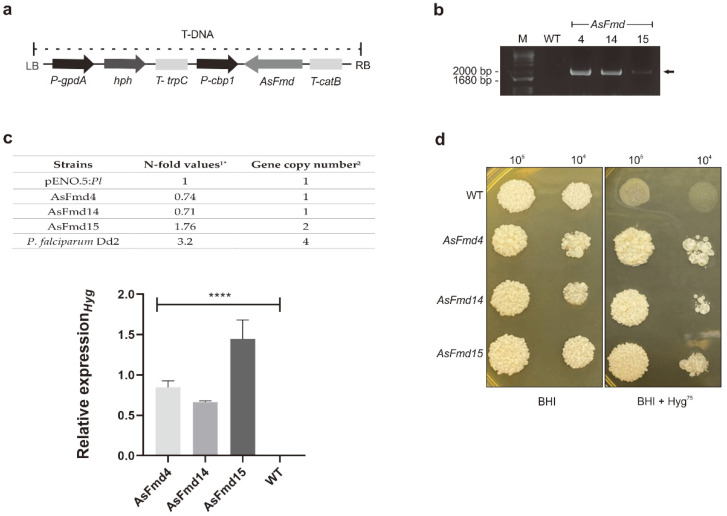
Confirmation of T-DNA integration into *P. lutzii* genome. (**a**) Schematic representation of the T-DNA generated for ATMT transformation. Antisense *fmd* fragment was constructed under control of calcium binding protein (*cbp1*) promoter of *H. capsulatum*, and *catB* terminator of *A. fumigatus*. Hygromycin-resistant gene (*hph*) was constructed under control of glyceraldehyde 3-phosphatate dehydrogenase (*gpdA)* promoter and trpC terminator of *A. nidulans*. (**b**) Confirmation of antisense plasmid integration on *P. lutzii* genome. Black arrow indicates the amplicon of 1800 bp on transformed strains. (**c**) Upper panel: Copy number of inserted Hygromycin into *P. lutzii* yeast cells accessed through qPCR. ^1^* N-Fold copy number of AsFmd silenced strains of *P. lutzii* in comparison to luciferase promotor gene pENO.5:Pl that harbors one copy of hygromycin gene [35] and *P. falciparum* Dd2 that contains 4 copy numbers of multidrug resistance protein *pfmdr1* gene [36]. N-fold of *P. lutzii* was achieved by the average of triplicate of hygromycin expression levels. Hygromycin Ct values was normalized with enolase reference gene (PAAG_11169). ^2^ Linear regression was used for estimating the gene copy number in unknown silenced strains in relation to *P. falciparum* Dd2 [36] that contains 4 copy numbers and pENO.5:*Pl* that harbors one copy of hygromycin gene. The regression statistic was R^2^: 0,98 and Χ variable with standard error 0.09 and *p* ˂ 0.0008. Lower panel: Relative quantification of hygromycin expression in WT and AsFmd strains through RT-qPCR. The enolase gene (PAAG_11169) was used as endogenous control. Shapiro–Wilk test was employed to determine data normality (*p* values > 0.1). Statistical analyses from experimental triplicates were performed through one-way ANOVA. **** represents *p* values < 0.0001. (**d**) WT and AsFmd strains growth in the absence (first panel) or presence of hygromycin (75 μg/mL) (second panel). Cells were plated at 10^5^ and 10^4^ cells/mL concentrations. Images were obtained after 7 days of growth at 37 °C.

**Figure 2 microorganisms-10-02011-f002:**
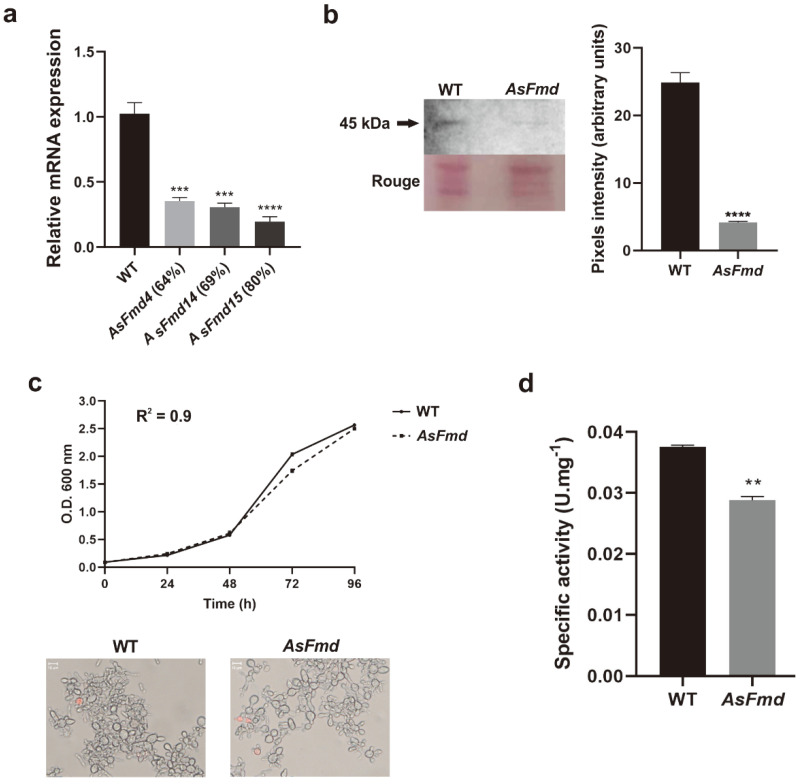
*P. lutzii fmd* gene silencing. (**a**): Relative quantification of *fmd* expression in WT and AsFmd strains through RT-qPCR. The tubulin gene (PAAG_03031) was used as endogenous control. (**b**): Fmd expression analysis in WT and AsFmd strains through immunoblotting. Rouge: WT and AsFmd protein extract stained with Rouge-Ponceau exhibiting similar protein quantification. Polyclonal antibody anti-*Pb*Fmd was incubated with 30 μg of protein extracts. Pixel intensity was measured by densitometric analysis of immunoblotting bands using ImageJ software. Pixel intensity from three replicates was measured by densitometric analysis of immunoblotting bands using ImageJ software. (**c**): Growth and viability of WT and AsFmd strains. WT and AsFmd strains were grown in BHI for 96 h. O.D. was measured daily at 600 nm to determine growth curve. Microscopy represents *P. lutzii* cell viability that was accessed by staining with propidium iodide (1 μg/mL) on the last day of the growth curve. Dead cells are colored in red. Images were obtained using an Axioscope A1 microscope (Carl Zeiss) at 493/623 nm and magnified 1000×. (**d**): Formamidase enzymatic activity in 500 ng of WT and AsFmd protein extract. Error bars represent standard deviation of three experimental replicates. Shapiro–Wilk test was employed to determine data normality: RT-qPCR (*p* values > 0.2), densitometric analysis of immunoblotting bands (*p* values > 0.2) and formamidase activity (*p* values > 0.8). Student’s t-test was applied for statistical analysis of relative quantification of *fmd* expression through RT-qPCR and densitometric analysis of immunoblotting bands. **** represents *p* values < 0.000, *** represents *p* values ≤ 0.0005 and ** represents *p* values ≤ 0.005.

**Figure 3 microorganisms-10-02011-f003:**
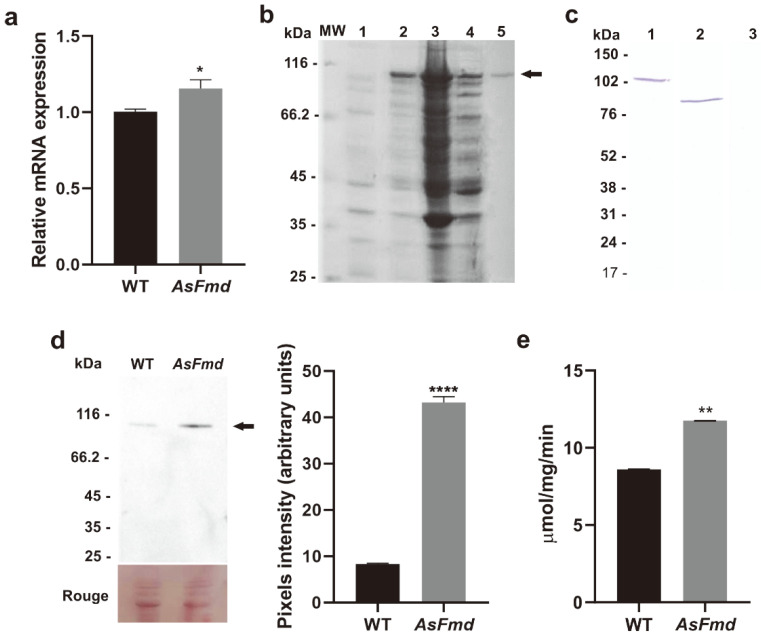
Urease expression, immunoblotting, and enzymatic activity analysis. (**a**): Relative quantification of urease expression in WT and AsFmd strains through RT-qPCR. The tubulin gene (PAAG_03031) was used as endogenous control. Statistical analyses were performed through Student’s *t*-test. * represents *p* values < 0.05. (**b**): Heterologous expression and purification of recombinant urease analysis through SDS-PAGE. MW: protein molecular weight marker. 1: *E. coli* protein extract prior to IPTG induction. 2: Urease expression (arrow, 103 kDa) after 3 h of induction with 1.0 mM IPTG. 3: Pellet after cell lysis. 4: A 103 kDa protein, equivalent to urease size, on soluble supernatant after cell lysis. 5: Purified recombinant urease after affinity chromatography with NI-NTA AGAROSE resin. (**c**): Immunoblotting for anti-Ure antibody specificity test. 1: Incubation of anti-Ure (1:5000) with 10 μg of purified rUre, rendering a 103 kDa protein, equivalent to 83 kDa of *Pb*Ure fused to 20 kDa of His-tag. 2: Incubation of anti-Ure (1:5000) with 30 μg of *P. lutzii* total protein extracts. 3: Incubation of pre-immune serum (1:500) with 30 μg of *P. lutzii* total protein extracts. (**d**): Analysis of urease expression in WT and AsFmd strains through immunoblotting. The arrow indicates urease protein (85 kDa). Rouge: WT and AsFmd protein extract stained with Rouge-Ponceau exhibiting similar protein quantification. Pixel intensity was measured by densitometric analysis of immunoblotting bands from experimental triplicates using ImageJ software. (**e**): Urease enzymatic activity in 500 ng of WT and AsFmd protein extract. Shapiro–Wilk test was employed to determine data normality: RT-qPCR (*p* values > 0.4), densitometric analysis of immunoblotting bands (*p* values > 0.5) and urease activity (*p* values > 0.6). Student’s t-test was applied for statistical analysis of RT-qPCR, densitometric analysis of immunoblotting bands and urease activity. **** represents *p* values < 0.0001 and ** represents *p* values ≤ 0.005. Error bars represent standard deviation of three experimental replicates.

**Figure 4 microorganisms-10-02011-f004:**
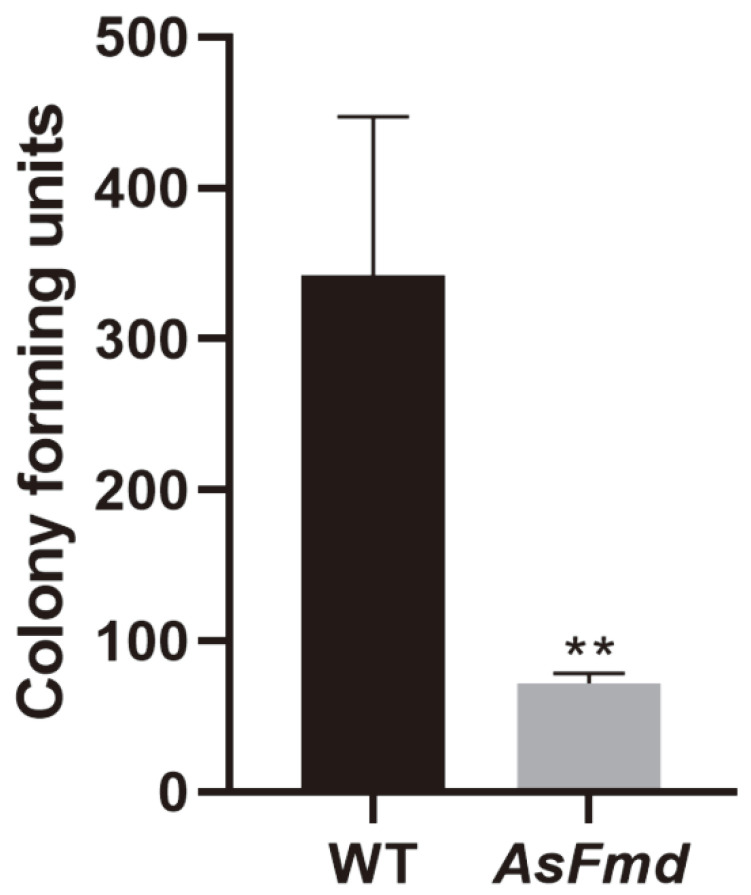
*P. lutzii* WT and AsFmd survival after macrophage infection. Following to recovering of ex vivo infection in murine J774 macrophages, CFU was counted at 7 days growth after the period of incubation. Shapiro–Wilk test was employed to determine data normality (*p* values > 0.2). Student’s *t*-test was performed on experimental triplicate from biological replicate data in order to understand the magnitude of existent differences between the CFU of the two groups. ** represents *p* values ≤ 0.005. Error bars represent standard deviation of two experimental replicates from biological duplicates.

## Data Availability

Not applicable.

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
