# Peer review of "Paracoccidioides lutzii Formamidase Contributes to Fungal Survival in Macrophages"

_microorganisms, 2022, doi:10.3390/microorganisms10102011_

Round 1

Reviewer 1 Report

The manuscript reports an interesting study that fits well in this journal. The aim is relevant, and the results add new information to our current knowledge of this organism. However, I have the following concerns:

Major concerns:
The authors failed in to demonstrate the number of insertional events of the binary plasmid within the fungal genome. The system already has the handicap that insertional events are random, and the localization of the insertional events within the genome may be technologically challenging. However, the assessment of the number of insertional events can be easily achieved by qPCR. The lack of this parameter undermines the relevance of the phenotypical characterization.
The experimental design requires the inclusion of the WT strain transformed with the empty binary vector for a proper comparison with the silenced strains. In a formal sense, the binary vector contains at least one gene that is expressed in the fungal cell, the molecular marker. What is the influence of this on the analyzed phenotypes? The proper comparison should be silenced mutants vs. WT+vector, not vs. WT strain. Otherwise, the comparison is biased.

The manuscript lacks a section on statistical analyses. I strongly suggest the authors apply non-parametric tests for data analysis.

The silenced strains showed moderate silencing but it is far from total silencing, was it not possible to find silenced strains with levels higher than the 90% gene silencing?

Why did the authors not assay these strains in vivo? It results obvious to perform such kinds of assays with cells where a putative virulence determinant is lost/downregulated as in this case. In vivo analyses should be included in the revised version of the manuscript.

Minor points:
There are sentences in the introduction (lines 104 and 125) where the statement is supported by unpublished data. I think these are relevant and should be included in this study, at least, as supplementary information.

Please include references to support the use of the tubulin coding gene as a control for data normalization.

Author Response

Please, find the attached file

Reviewer 2 Report

This article aims to evaluate the formamidase function in Paracoccidioides lutzii by inhibiting the protein production using antisense mRNA. The autors describe an increase of urease production and activity in case of formamidase depletion and a reduced survival with macrophages in case of formamidase depletion.

The proposed methodology using antisense mRNA and the conclusions concerning urease and the survival of this fungus inside the macrophages are interesting. However, there are some questions on the methodology, especially on the number of biological and technical replicates, the choice of BHI and the survival of macrophages.

Major comments :

- The number of replicates in the different experiments is not detailed, either in Materials and methods, nor under figures, apart from "biological triplicates" in 2.5 and only 2 biological replicates in 2.11. Moreover, the technical replicates are not detailed, especially for RTqPCR.

- In part 2.5 : quantity of mRNA and proteins are reduced but there is no complementary fonctional test. The remaining presence of transcript could be sufficient to perform most of formamidase required activity in condition of nitrogen starvation.

- In part 2.6 and 2.8 : it is said that culture was performed in BHI. Is it a medium containing only secondary nitrogen sources (no ammonia, glutamine or glutamate)? If not, this would not be the best medium to evaluate formamidase depletion.

- In part 2.11 : there is no detail on macrophage survival : that could be washed off and that could give information on the virulence of WT and mutated strain.

Minor comments :

- l.106-107 : I don't see a proof that it is AreA in the given reference? Maybe use of "GATA family transcription factor suche as AreA" or add the fact that it is a prediction?

- Multiple use of the word "once" (lines 33, 79,  355, 472) for what seems to be a causality link link.

-  line 103, The use of "NCR conditions" is confusing : Do you mean "nitrogen starvation?". It could mean the condition of NCR mechanism repression which would be the case of nitrogen preferential sources abundance?

Reviewer 3 Report

The article shows how the authors developed a P lutzii silenced strain of fmd gene to demonstrate if formamidase contributes to the pathogen's survival in macrophages. The authors also indicate that this silenced strain expresses urease as a possible alternative route that serves as a nitrogen source.

The article is well written. The methods are widely described, and the results are clearly stated. The figures adequately illustrate the results. The discussion and conclusions are well focused on the results obtained.

As an objection, I have to comment that the authors cite "unpublished data" (lines 125, 424, 456) and "data not shown" (line 357) on multiple occasions. Authors should minimize this type of citation in the manuscript.

On the other hand, line 69 ("their function on fungal pathogenesis are reviewed by SANTOS et al., 2020") lacks the numerical citation among those included in the References section.

Author Response

To: Reviewer 3

Goiania, September 8th, 2022

Dear Reviewer 3,

Thank you for your comments, we appreciate them.

Please, find below the Response Letter:

Reviewer’s comments, questions and Answers

Reviewer 3: The article shows how the authors developed a P. lutzii silenced strain of fmd gene to demonstrate if formamidase contributes to the pathogen's survival in macrophages. The authors also indicate that this silenced strain expresses urease as a possible alternative route that serves as a nitrogen source.

The article is well written. The methods are widely described, and the results are clearly stated. The figures adequately illustrate the results. The discussion and conclusions are well focused on the results obtained.

Comment 1: As an objection, I have to comment that the authors cite "unpublished data" (lines 125, 424, 456) and "data not shown"(line 357) on multiple occasions. Authors should minimize this type of citation in the manuscript.

Answer: Thank you for your comment. These “unpublished data” are part of another work from our group that is being finalized and we will publish it soon.

Comment 2: On the other hand, line 69 ("their function on fungal pathogenesis are reviewed by SANTOS et al., 2020") lacks the numerical citation among those included in the References section

Answer: We apologize for this mistake. We corrected it and included the numerical citation for this reference in the manuscript (line 68).

Thank you for your contribution in improving our manuscript.

We hope to hear from you soon

Sincerely yours,

Clayton Luiz Borges, PhD

Laboratório de Biologia Molecular

Instituto de Ciências Biológicas

Universidade Federal de Goiás,

74001-970, Goiânia, Goiás, Brazil.

Tel:+55-62-35211110

e-mail:< clayton@ufg.br>

Round 2

Reviewer 1 Report

The authors addressed most of my original comments, thank you. I understand that some data cannot be disclosed because are part of an ongoing independent publishing process.

Reviewer 2 Report

I thank the authors for their answers to my comments. I still see three main flaws :

- the activity of formamidase is only reduced of about 25% compared to WT strain. 

- There is no evaluation in nitrogen starvation conditions, although formamidase contribution to nitrogen acquisition is mainly described during nitrogen starvation. Same thing for urease evaluation (which is a secondary nitrogen metabolism).

- The number of biological replicates is low, especially considering the variability observed in the macrophages experiments (on WT strain).